# Evaluation of Mitochondrial Function in Blood Samples Shows Distinct Patterns in Subjects with Thyroid Carcinoma from Those with Hyperplasia

**DOI:** 10.3390/ijms24076453

**Published:** 2023-03-29

**Authors:** Julia Bernal-Tirapo, María Teresa Bayo Jiménez, Pedro Yuste-García, Isabel Cordova, Ana Peñas, Francisco-Javier García-Borda, Cesar Quintela, Ignacio Prieto, Cristina Sánchez-Ramos, Eduardo Ferrero-Herrero, María Monsalve

**Affiliations:** 1Hospital Universitario 12 de Octubre, Av. Cordoba Km. 5.4, 28041 Madrid, Spain; juliab01@ucm.es (J.B.-T.); pyuste@ucm.es (P.Y.-G.); fjgarc01@pdi.ucm.es (F.-J.G.-B.); eduardo.ferrero@salud.madrid.org (E.F.-H.); 2Instituto de Investigaciones Biomédicas “Alberto Sols” (CSIC-UAM), Arturo Duperier 4, 28029 Madrid, Spain; maite.bayo@ugr.es (M.T.B.J.); isabel_cordova@iislafe.es (I.C.); anappdv@gmail.com (A.P.); nprieto58@gmail.com (I.P.); cristina.sanchez@cnic.es (C.S.-R.)

**Keywords:** metabolism, mitochondria, cancer, thyroid, diagnostic

## Abstract

Metabolic adaptations are a hallmark of cancer and may be exploited to develop novel diagnostic and therapeutic tools. Only about 50% of the patients who undergo thyroidectomy due to suspicion of thyroid cancer actually have the disease, highlighting the diagnostic limitations of current tools. We explored the possibility of using non-invasive blood tests to accurately diagnose thyroid cancer. We analyzed blood and thyroid tissue samples from two independent cohorts of patients undergoing thyroidectomy at the Hospital Universitario 12 de Octubre (Madrid, Spain). As expected, histological comparisons of thyroid cancer and hyperplasia revealed higher proliferation and apoptotic rates and enhanced vascular alterations in the former. Notably, they also revealed increased levels of membrane-bound phosphorylated AKT, suggestive of enhanced glycolysis, and alterations in mitochondrial sub-cellular distribution. Both characteristics are common metabolic adaptations in primary tumors. These data together with reduced mtDNA copy number and elevated levels of the mitochondrial antioxidant PRX3 in cancer tissue samples suggest the presence of mitochondrial oxidative stress. In plasma, cancer patients showed higher levels of cfDNA and mtDNA. Of note, mtDNA plasma levels inversely correlated with those in the tissue, suggesting that higher death rates were linked to lower mtDNA copy number. In PBMCs, cancer patients showed higher levels of PGC-1α, a positive regulator of mitochondrial function, but this increase was not associated with a corresponding induction of its target genes, suggesting a reduced activity in cancer patients. We also observed a significant difference in the PRDX3/PFKFB3 correlation at the gene expression level, between carcinoma and hyperplasia patients, also indicative of increased systemic metabolic stress in cancer patients. The correlation of mtDNA levels in tissue and PBMCs further stressed the interconnection between systemic and tumor metabolism. Evaluation of the mitochondrial gene *ND1* in plasma, PBMCs and tissue samples, suggested that it could be a good biomarker for systemic oxidative metabolism, with *ND1*/mtDNA ratio positively correlating in PBMCs and tissue samples. In contrast, *ND4* evaluation would be informative of tumor development, with *ND4*/mtDNA ratio specifically altered in the tumor context. Taken together, our data suggest that metabolic dysregulation in thyroid cancer can be monitored accurately in blood samples and might be exploited for the accurate discrimination of cancer from hyperplasia.

## 1. Introduction

Metabolic adaptations are a hallmark of cancer and to some extent recapitulate the changes in metabolism that take place during the cell cycle [1]. Eukaryotic cell proliferation requires that cells switch from mitochondrial oxidative metabolism to glycolysis as the main source of ATP [2]. Growth factors are important drivers of this metabolic change, whereas mediators that signal for conditions of low nutrient availability induce cellular quiescence and activate oxidative metabolism [3,4]. Cancer cells typically make an altered use of mitochondria, which can result in a diminished capacity to synthesize ATP, but generally associates with changes at the level of the tricarboxylic acid cycle (TCA) such as reduced succinate dehydrogenase (SDH) activity [5]. In turn, this is linked to an increased avidity for glutamine [6] and to the activation of anaplerotic reactions that aim to compensate for TCA deficits by replenishing the depleted pools of key metabolic intermediates such as oxaloacetate [7]. The “dysfunctional” mitochondria of cancer cells also generally produce elevated levels of reactive oxygen species (ROS), which can both enhance cell proliferation [8] and compromise cellular viability [9]. Indeed, a considerable number of chemotherapeutic agents exploit this characteristic of cancer cells [10]. Enhanced ROS production can result in damage to mitochondrial DNA (mtDNA) [11], and a specific pattern of damaged mtDNA molecules has been found associated with different types of cancer [12]. Normal cells circumvent the accumulation of oxidized/mutated mtDNA by activating different repair and degradation mechanisms, which tend to be insufficiently activated in cancer cells [13]. Uncontrolled cell growth in tumors generally boosts tumor cells metabolic demands and can override the metabolic supply [14]. Accordingly, the preservation of metabolic plasticity provided by mitochondria is also fundamental for cancer cell survival [15]. The inability to react to metabolic strain can lead to cell death, and dying cancer cells shed their mutated genomic and mtDNA into the blood stream, which can be non-invasively and accurately evaluated by PCR analysis [16]. The potential applicability of this type of analysis of mtDNA mutations for diagnostic and prognostic evaluation of patients with cancer is under active investigation [17]. However, its application to the clinical setting has so far been successful in a very limited number of cases [18].

Thyroid cancer is the most common endocrine cancer, and the eleventh most common cancer type. The incidence of thyroid cancer has been steadily increasing by about 5% each year over the last 10 years, more rapidly than any other cancer type. The overall incidence of thyroid cancer has increased in people of all racial/ethnic groups and in both males and females, although incidence is nearly three times higher in women than in men [19]. At least part of this increase is attributable to improved detection methods.

A major risk factor for thyroid cancer is exposure of the head and neck to radiation, especially during childhood. Importantly, recent studies have also described an elevated incidence of thyroid and other cancer types associated with metabolic dysfunction (MetS), with metabolic derangements also having a strong impact on progression and overall mortality rates, reviewed in [20]. It is therefore possible that an analysis of the systemic metabolic state of a patient might have prognostic value for thyroid cancer [21].

Common diagnostic procedures for thyroid cancer include thyroid echography and needle puncture biopsy of thyroid nodules for histological evaluation. Typically, people that have suspicious thyroid nodules undergo total or partial thyroidectomy; however, about 50% of suspicious cases show hyperplasia and not carcinoma. The heavy socio-economic burden this creates requires the development of novel and cost-effective diagnostic tools to deliver the best standards of care to patients. In this line, improved imaging techniques and identification of non-invasive biomarkers in blood and other biological fluids are the focus of intense research [22].

Pre-surgical detection of somatic mutations in fine-needle aspiration biopsy (FNAB) has proven to have significant predictive value in thyroid cancer, but the low positive predictive value of some tests and the high costs involved have greatly restricted their implementation in routine clinical practice [23]. Interestingly, an evaluation of cellular mtDNA content in thyroid tumors found increased mtDNA levels in Hürthle Cell Carcinoma (HCC), which was characterized by high numbers of abnormal mitochondria, leading the authors to propose that its evaluation, along with other molecular biomarkers, might have utility in the pre-operative differential diagnosis of thyroid nodules [24]. A recent study also evaluated whether changes in total mtDNA copy number are related to systemic metabolic changes and whether they can be detected in leukocytes, finding that increased mtDNA copy number in PBMCs might be a risk factor for thyroid cancer [25]. It has been suggested that these increases represent a compensatory response of mitochondrial biogenesis related to increased ROS production and high mutation rates [26]. In fact, thyroid tumors with Hürthle cells also show increased levels of mtDNA mutations and deletions [27]. In fact, some mtDNA haplotypes and SNPs have been associated with thyroid cancer, and several mtDNA mutations identified in thyroid tumors have been demonstrated to be pathogenic [28]. The possibility of using the analysis of cell-free DNA (cfDNA) in plasma samples to diagnose thyroid tumors has also been tested, and a recent study has suggested, that the ratio of two mitochondrial genes, *ND4* and *ND1* in plasma samples is increased in patients with thyroid cancer [29]. Moreover, altered levels of nuclear-encoded mitochondrial proteins have also been proposed as a biomarker for thyroid cancer [30], further stressing the relevance of mitochondrial function in this context.

We examined the metabolic adaptations in thyroid cancer and their correlation with biomarkers of both systemic and tumor-associated metabolic dysfunction in blood samples, in an attempt to better characterize the metabolic characteristics that define thyroid cancer and identify biomarkers with diagnostic value. We found that patients with thyroid tumors present an altered mitochondrial oxidative stress signature detectable in the tissue and in peripheral blood mononuclear cells (PBMCs) that impacts on the presence of mtDNA fragments detectable in plasma samples. These characteristics can be used to discriminate hyperplasia from carcinoma using minimally invasive procedures.

## 2. Results

### 2.1. Characterization of the Cohort: Basic Metabolic Risk Factors and Histological Analysis

A cohort of 45 patients undergoing thyroidectomy because of suspicion of thyroid cancer was recruited at the Hospital Universitario 12 de Octubre, (H12O, Madrid, Spain); of those, 24 patients were subsequently diagnosed with non-malignant hyperplasia by the Pathological Anatomy Unit and 21 were diagnosed with malignant thyroid tumors. In a second cohort of 28 subjects, 16 patients were diagnosed with hyperplasia and 12 with carcinoma. The distribution of the patients between the hyperplasia and carcinoma groups was not significantly different based on sex; however, in cohort 1 the hyperplasia group was significantly older than the carcinoma group (Table 1).

We determined whether differences in common metabolic risk factors could be identified between the groups. No significant differences were found for body mass index, fasting glucose, total cholesterol and triglycerides between the groups in either cohort (Table 1). Likewise, blood pressure and smoking rates were similar between the groups in both cohorts (Table 1). These findings are in line with previous studies suggesting that while metabolic dysfunction is a risk factor for thyroid cancer, the metabolic parameters commonly evaluated in clinical practice are not suitable diagnostic tools [31]. A recent epidemiological study based in Korea with 173,343 participants showed that metabolically unhealthy women, either normal weight or obese, had an increased risk of thyroid cancer [HR (95% CI) = 1.57 (1.02–2.40) and 1.71 (1.21–2.41), respectively) compared with non-obese women without metabolic abnormalities. However, this significant association was not observed in men. Thyroid cancer risk was higher among non-obese women with high Waist Circumference (WC) [≥85 cm; HR (95% CI) = 1.62 (1.03–2.56)] than in non-obese women with low WC, and in obese women with low HDL-cholesterol [<50 mg/dL; HR (95% CI) = 1.75 (1.26–2.42)] compared with non-obese women with high HDL-cholesterol [32].

We next examined the histology of the thyroid samples from Cohort 1. Structural alterations are generally more evident in cancer tissue; however, a preliminary evaluation of the non-fixed samples obtained shortly after surgery correctly discriminated only about 2/3 of the samples. When possible, in addition to the nodule, a peripheral tissue sample was also taken and analyzed. Representative H&E-stained images of peripheral and central tissue of two samples classified as hyperplasic during the preliminary evaluation are shown in Figure 1A. One of the samples was subsequently found to be a carcinoma. As shown in the images, low-level structural alterations can also be clearly observed in the peripheral tissue samples and the degree of alterations in some hyperplastic tissues can make an accurate diagnosis challenging. Nevertheless, the most common histological staining used by pathologists for thyroid cancer diagnosis is still H&E staining [33].

We next determined the number of proliferating cells in thyroid samples by immunochemical staining with an antibody against Ki67 [34]. As expected, we found significant differences in the number of proliferating cells between peripheral and central tissue samples; however, no significant differences were found in Ki67-positive cells between carcinoma and hyperplasia samples, although there was a trend for a greater number of Ki67-positive cells in the former in the two cohorts (Figure 1B). We then evaluated apoptosis in cryo-sections of tissue samples from cohort 1 using a TUNEL assay. Nuclei where labeled with DAPI and apoptotic nuclei were considered as those positive for both TUNEL and DAPI staining [35] As expected, cancer samples had more positive cells than hyperplasia samples (Figure 1C), but the differences were not significant, likely due to the small sample size. These observations, consistent with published data, also highlight the difficulties of a clear-cut diagnosis prior to surgery and in the absence of a molecular signature.

As dead tumor cells commonly fail to properly degrade their gDNA, increased levels of shed cfDNA are frequently found in the blood of patients with cancer and, accordingly, its quantification has been proposed as a biomarker in some cases [36]. We measured the levels of gDNA in plasma samples of cohort 1, finding significantly higher levels in patients with cancer than in peers with hyperplasia (Figure 1C), an observation also consistent with previous findings in other cancer types [37], supporting the presence of elevated cell death rates in the cancer group. Overall, our results show that proliferation and apoptotic activity are higher in thyroid cancer than in equivalent hyperplasia samples, but their determination is likely not sufficiently robust to discriminate between the two entities using standard procedures.

A more recently characterized feature of cancer is the presence of an abnormal vascular tree [38]. Cancer induces angiogenesis, but the tumor vasculature is generally tortuous and poorly perfused, making of it one of the hallmarks of cancer. We thus analyzed the vascular tree in thyroid tissue samples using antibodies directed against SMA, to label the large vessels and measure the regularity of the blood vessels, and against VEGFR2, a protein receptor present in endothelial cells and induced in response to pro-angiogenic factors, to examine the induction of angiogenesis [39]. In line with previous works on other cancer types [40], both VEGFR2 levels and vascular tortuosity were significantly greater in cancer samples than in hyperplasia samples, despite the small sample size (Figure 1D), suggesting that vascular abnormalities are a more robust feature of cancer than classical markers. Supporting this finding, western blotting of VEGFR2 in whole tissue extracts revealed significantly elevated VEGFR2 activity in cancer samples, as determined by the ratio of phosphorylated (p)-VEGFR2/VEGFR2 (Figure A1). Notably, this is a feature which, has been shown to be modulated by metabolic regulators such as PGC-1α [41].

### 2.2. Characterization of Thyroid Tissue Metabolic Status

As noted above, while cancer cells display several metabolic traits, mitochondrial activity is particularly affected. Indeed, the ratio of anaerobic-to-aerobic metabolism is generally high, as cancer cells prefer to generate ATP from glucose rather than from oxygen. ATP-coupled oxygen consumption is often compromised, which is frequently linked to mitochondrial dysfunction and elevated ROS production [42]. The resulting poor coupling capacity of the electron transport chain (ETC) leads to enhanced H^+^ leakage and O_2_^-^ generation, both common characteristics of mitochondrial oxidative stress. The poor mitochondrial performance is normally compensated for by an increase in glycolytic flux [34] and by a partially effective compensatory induction of antioxidant systems [43]. Many cancer cell types are actually reliant on mitochondrial activity, which can fuel tumor growth and survival [44]. Thyroid cancer, in particular, has been shown to have in some cases large amounts of active mitochondria [45].

To evaluate the metabolic status of the thyroid samples, we first assessed the number and subcellular distribution of mitochondria in fixed tissue sections by immunofluorescence (IF) using a specific antibody against the mitochondrial protein TOMM22 [46]. Consistent with data for total mitochondrial content in many tumors [46,47], total TOMM22 positive area per cell was highly variable and did not correlate with disease state (Figure A2), and was therefore disregarded for subsequent analysis. Instead, we used TOMM22 IF signal to monitor two parameters indicative of mitochondrial dysfunction: the asymmetric distribution of mitochondria around the nucleus (*capping*) and the degree of mitochondrial fission [48,49]. Both were evaluated by quantitative analysis of cellular cross-section signal profiles. *Capping* was defined as the ratio of signals at the two ends of the nucleus periphery, which was described by the DAPI signal. The fission rate was defined as the standard deviation of the signal across the cytosol. While the two parameters were found increased in cancer tissues in both cohorts, only the differences in *capping* reached statistical significance (Figure 2A).

Poor mitochondrial function is commonly associated with an increased reliance on glucose. Enhanced activation of the kinase AKT is a common feature of thyroid cancer [50] and, importantly, AKT activation plays a fundamental role in the upregulation of glucose uptake [51]. We monitored AKT activation in thyroid tissue by immunofluorescence using a specific antibody against its phosphorylated form. Results showed that p-AKT levels were significantly higher in cancer tissue samples than in hyperplasias (Figure 2B), suggesting that poor mitochondrial function is associated with elevated glycolytic flux in thyroid cancer a finding consistent with previous reports [52].

Given that mitochondrial dysfunction is typically associated with mitochondrial oxidative stress, we measured the levels of PRX3, a mitochondrial antioxidant protein induced by oxidative stress [53] and elevated in various cancers, and a target gene of the oncogene *MYC* [54]. Western blotting revealed that PRX3 levels were significantly higher in carcinomas than in hyperplasias (Figure 2C, left panel), suggesting increased mitochondrial oxidative stress. Mitochondrial oxidative stress can result in oxidative damage to mtDNA, deficient mtDNA replication and gradual loss of mtDNA [55]. Indeed, reduced levels of mtDNA have been found associated with some cancer types [47]. We analyzed mtDNA levels in tissue samples by qPCR and found that mtDNA levels were significantly lower in carcinomas than in hyperplasias (Figure 2C, right panel), suggesting loss of mtDNA copy number. Overall, the data suggest that thyroid carcinoma tissues display characteristics consistent with the presence of mitochondrial dysfunction and mitochondrial oxidative stress.

### 2.3. Evaluation of Thyroid DNA in Plasma Samples

Enhanced rates of cell death are a common characteristic of carcinomas, and tend to result in the incomplete degradation of DNA, which can be found in the blood [56]. Evaluation of the levels and DNA mutations present in circulating cfDNA forms the basis of several liquid biopsy protocols for the non-invasive diagnosis and stratification of cancer [57]. When we measured the levels of gDNA in plasma samples, we consistently found them significantly elevated in the carcinoma group (Figure 2D, left panel). In contrast, mtDNA levels in plasma were similar in both groups. Nevertheless, we found a significant and positive correlation between the levels of mtDNA and gDNA in the plasma samples of the subjects analyzed, suggesting that both had a common origin (Figure 2D, right panel). Of note, when we compared the levels of mtDNA in plasma and thyroid tissue, we found an inverse correlation (Figure 2E, top panel), possibly suggesting that the cells that were more likely to die and release mtDNA into the blood were those that had less mtDNA. This could thus generally relate the rate of cell death with mitochondrial dysfunction, but more likely it could indicate that the cells that died in a non-programmed manner with incomplete DNA degradation were those with higher mitochondrial dysfunction. Definite conclusions cannot be drawn since the contribution of live cells to the process could not be evaluated.

To further investigate the mechanisms involved, we compared mtDNA/gDNA ratios between plasma and tissue samples in hyperplasia and carcinoma groups. We consistently found a significant inverse correlation in carcinoma samples, but no correlation between the two values in hyperplasia samples (Figure 2E, bottom panels), possibly suggesting that the enhanced release of mtDNA into the plasma associated with mitochondrial dysfunction, dependent or independent of cell death, was a feature particularly prevalent in patients with carcinoma. Still, a caveat to this analysis is that the assay does not provide formal proof of the source of the identified mtDNA nor gDNA, although the results support the presence of mitochondrial dysfunction in cancer samples, in line with previous findings.

### 2.4. Evaluation of the Systemic Metabolic Status of Patient-Derived PBMCs

We next aimed to analyze the general metabolic status of the patients to examine for a correlation between general metabolic status and disease state. To do this, we used PBMCs, although we were aware of the unavoidable confounding effect of the inflammatory state of the patients, which might activate immune cell metabolism. We used western blotting to assess the protein levels of PGC-1α, a master positive regulator of oxidative metabolism and mitochondrial biogenesis [58], PRX3, a mitochondrial antioxidant and gene target of both PGC-1α [59] and c-MYC [54]; and PFKFB3, a positive regulator of the glycolytic flux that is also tightly regulated at the transcriptional level [60]. We found that while the steady-state levels of all three proteins were higher in PBMCs from carcinoma samples than from hyperplasia samples, only PGC-1α levels were significantly greater (Figure 3A). Given these findings, we analyzed the gene expression of these proteins and several target genes of PGC-1α [59]: two components of the ETC, *ATP5BP (*coding for the ATP Synthase subunit ß1) and *COXIV* (coding for Cytochrome Oxidase subunit IV); a mitochondrial antioxidant, *SOD2 (*coding for Mn Superoxide Dismutase); and a regulator of mitochondrial replication and gene expression, *TFAM* (coding for Mitochondrial Transcription Factor A). Results showed that only *PGC1A* expression was significantly higher in the carcinoma group than in the hyperplasia group (Figure 3B, top panel), possibly suggesting that the increase in *PGC1A* expression was related to the pro-inflammatory state did not have the expected functional impact on PBMCs. To evaluate this possibility, we compared the goodness of the correlation of the expression of the tested genes in the hyperplasia and carcinoma groups. We found a significant difference for the *PFKFB3*/*PRDX3* correlation, with *PFKFB3/PRDX3* values higher in the carcinoma group (Figure 3B, bottom panel). We also observed that *PFKFB3* and *PGC1A* values correlated significantly and positively only in carcinoma samples (Figure 3B, bottom panel); however, the differences were not significant between the groups (*p* = 0.18). Altogether, these observations suggest that *PGC1A* is more strongly induced in PBMCs of cancer patients, but its activity does not seem to be increased at the same level. Furthermore, although no significant increases in the glycolysis regulator *PFKFB3* nor in antioxidant genes such as *PRDX3* could be observed in the carcinoma group, the observed differences in the ratios of *PFKFB3/PRDX3* and *PFKFB3/PGC1A* might be functionally relevant, stressing the reduction in PGC-1α activity, and possibly in the antioxidant capacity in carcinoma patients. As a further test for PGC-1α activity, we evaluated the mtDNA content of PBMCs and found that levels were, on average, lower in the carcinoma group than in the hyperplasia group (Figure 3C), although the differences were not significant (*p* = 0.08). This finding is in line with the possible reduction of PGC-1α activity in PBMCs of cancer patients, despite the increase in mRNA and protein levels which could lead to mitochondrial dysfunction.

### 2.5. Analysis of the Systemic Inflammatory Status

Tumor development is associated with an immune response, and a general systemic pro-inflammatory state that, in turn, affects both systemic and cancer metabolism [61]. Therefore, we measured the levels of several important inflammatory mediators in plasma samples by multiplex flux cytometry. Of note, no significant differences were found in the levels of IL-1ß, IL-4, IL-6, IL-10 and TNFα between hyperplasia and carcinoma groups (Figure 4A). We also found no changes in the gene expression of *IL1B* and *IL4* (Figure 4B), detectable in PBMCs. Overall, these results suggest that the systemic pro-inflammatory status is similar in patients with thyroid carcinoma or hyperplasia, possibly suggesting that, the detectable metabolic differences in PBMCs cannot be solely attributed to the inflammatory status of the patients.

Inflammation, by activating NOX, and mitochondrial dysfunction increase mitochondrial superoxide production, which is commonly associated with oxidative damage to macromolecules. We thus analyzed the oxidative modification of DNA isolated from plasma samples using ELISA-based detection of 8-OhdG, finding that 8-OHdG levels once corrected for the total DNA load, were significantly lower in carcinoma than in hyperplasia samples. This observation was unexpected since tumor development is generally associated with increased oxidative stress (Figure 4C). As these differences could be related to the levels of antioxidants, we tested the total, fast and slow antioxidant capacity in plasma samples, but failed to find significant differences between the groups (Figure 4D). These results suggest that, either there is less oxidative stress in cancer patients or, the DNA released is less oxidized, possibly due to a decrease in DNA repair mechanisms.

### 2.6. Comparison of Systemic and Tumor Metabolism

Our results thus far suggest a possible correlation between the metabolic status of the tumor and that of the patient. To test this idea more directly, we first evaluated the correlation between the mtDNA copy number in PBMCs and in tumor tissues, and observed a positive and significant correlation between the two (Figure 5A, left panel), supporting a link between systemic and tumor metabolism. We then investigated whether this correlation changed with disease state, and found that this significant correlation was observed in patients with thyroid hyperplasia, but not in those with carcinoma (Figure 5A, right panel), possibly suggesting that this correlation was at least partially lost in carcinoma patients. However, the very small sample size limited the potential significance (*p* = 0.48) of this observation.

Cancer-related changes in metabolism can be associated with the accumulation of mtDNA molecules bearing deletions in mitochondrial genes. As a surrogate marker of mtDNA damage we assessed two mitochondrial genes: *ND1* and *ND4*. Mutations in both have strong a functional impact and pathological consequences. *ND4* gene deletions are a common finding in cancer while *NDI* deletions are more rare [55]. Targeting sequences detectable in plasma samples, we found that *ND1* levels correlated significantly and positively with the levels of mtDNA in both tumor tissues and PBMCs (Figure 5B), and the differences between them were not statistically significant (Figure 5B), an observation that stresses the genomic stability of *ND1*. In contrast, *ND4* levels correlated significantly and positively with total mtDNA in PBMCs only, not in tumor samples (Figure 5C), suggesting that also in the context of thyroid cancer *ND4* is less preserved. Indeed, analysis of *ND4* vs. mtDNA correlation showed significant differences between tumor tissue samples and PBMCs (Figure 5C), stressing that tumor samples show alterations in *ND4* copy number. Of note, loss of *ND4* has been generally associated with mitochondrial dysfunction and oxidative stress.

To further test this notion, and determine if the instability of *ND4* was restricted to the tumor tissue, we compared *ND1*/mtDNA and *ND4/*mtDNA ratios in PBMCs vs. tumor tissue samples. We found a significant positive correlation for *ND1* (Figure 5D), indicating that the stability of *ND1* is equivalent in the tumor and PBMCs and, thus, systemic. In contrast, we observed a significant negative correlation for *ND4/* mtDNA ratios between tissue and PBMCs (Figure 5E).

### 2.7. Origin and Significance of cfDNA Detectable in Crude Plasma Samples

The success of a biomarker not only depends on its sensitivity and specificity, but also on the cost and feasibility of the analysis. mtDNA has a strong potential as a non-invasive biomarker as it is a stable molecule that can be easily isolated from samples, especially plasma [62]. To determine whether the evaluation of mtDNA in plasma could be used to gauge the metabolic profile of thyroid cancer, we compared the overall plasma levels of *ND1* and *ND4* and mtDNA and/or gDNA ratios with those found in PBMCs or tumor tissue. Regarding *ND1*, we found that the ratio *ND1*/mtDNA/gDNA was inversely correlated in plasma vs. tissue samples, possibly suggesting that cells bearing stable mtDNA were less likely to shed mtDNA into the plasma. Analysis of *ND4* showed that only the total values found in plasma were significantly and positively related to those found in PBMCs (Figure 6B), but this correlation was lost when normalized for mtDNA content (Figure 6C), suggesting that the plasma ND4/mtDNA ratios were not indicative of the structure of the mtDNA present in PBMCs. All together these data suggest that testing of both *ND1* and *ND4* in crude plasma can provide information on tumor development.

Aiming to identify the origin of these differences, we next directly compared *ND1*/gDNA and *ND4*/gDNA ratios in tissue samples, PBMCs and crude plasma samples. We found that *ND1*/gDNA levels were significantly lower in carcinoma tissue than in hyperplasia tissue, consistent with a depletion of mtDNA in cancer tissue, whereas *ND4*/gDNA levels were significantly higher in carcinomas, suggesting an enrichment in carcinomas in *ND4* bearing mtDNA (Figure 6D). We next tested the *ND4/ND1* ratio. Again, we found a significant difference in this ratio between carcinoma and hyperplasia tissue samples, with carcinomas showing higher levels (Figure 6G). However, these differences were not observed in PBMCs or in crude plasma samples (Figure 6E,F). In fact, a trend for lower *ND1*/gDNA and *ND4*/gDNA ratios in carcinoma patients was noted in both PBMCs and plasma samples, in line with previous data suggesting that carcinoma patients have significantly more gDNA and on average, less mtDNA in plasma samples (Figure 2D). These data support the impact of tumor metabolism on tumor development and highlighted the potential use of the determination of *ND1* and *ND4* ratios, at least in tumor samples.

To elucidate the basis of the observed differences between tissue, plasma and PBMCs, we next tested the correlation of *ND1* vs. *ND4* in tissue, PBMCs and plasma samples. Notably, in patients with hyperplasia we found a positive and significant correlation between *ND1* and *ND4* in PBMCs and plasma samples, and a similar, but not significant, trend in tissue samples. In carcinoma patients the linear regression slope was significantly reduced in PBMCs and a similar trend was observed in tissue and plasma samples (*p* = 0.059), suggesting that *ND4* instability affected not only the thyroid tissue and could in fact be evaluated in plasma samples. This data also indicated that the determination of the *ND4/ND1* ratio in PBMCs or even in plasma, could in fact have a diagnostic value when these values are interpolated in a linear regression model.

To evaluate the generalization of this findings for *ND4* we tested the *ND4/ND1* ratio using standard *ND4* (*ND4s*) sequences but, in this case, we did not find significant differences among the groups (Figure A3).

### 2.8. Functional Significance of the Observed Changes in ND1 and ND4

We assessed the functional significance of the changes in *ND1* and *ND4* by comparing their levels in PBMCs and tissue samples against the changes in gene expression of metabolic-, inflammatory- and angiogenesis-related genes observed in PBMCs. Regarding metabolism, we found that both *TFAM/PRDX3* and *PFKFB3/PRDX3* ratios correlated negatively and significantly with *ND4*/mtDNA in the thyroid tissue (Figure 7A,C). In contrast, the *ATP5BP/PRDX3* ratio correlated positively and significantly with *ND1*/mtDNA in tissue (Figure 7B). These results suggested that there is in fact a functional correlation between the stability of the mitochondrial genome in the thyroid tissue and the systemic redox balance, detectable in PBMCs.

Next, we tested if the functional impact of these correlations extended to the immune response. Consistently, the analysis of inflammatory genes showed that *IL4,* an M2 biomarker*,* correlated negatively and significantly with *ND4*/mtDNA ratio in thyroid tissue, a correlation that was not observed for *IL1B,* an M1 biomarker (Figure 7D). Conversely, but not in PBMCs, there were significant differences in correlation between the two (Figure 7E). Finally, we tested the angiogenesis regulator *DLL4.* We found that it correlated negatively and significantly with *ND4*/mtDNA in tissues (Figure 7E). Altogether, these data support a functional link between the observed alterations in mtDNA in tumor samples, with gene expression changes in metabolism, inflammation and angiogenesis genes in PBMCs.

This functional link prompted us to investigate if the observed differences in *PGC1A* expression levels in PBMCs between patients with thyroid carcinoma and hyperplasias (Figure 3) could be also related to mtDNA stability. To this end, we grouped the subjects as high and low for *PGC1A* gene expression in PBMCs (Figure 7G). Consistent with the observed global gene expression differences, the number of patients with carcinoma was found to be significantly higher in the high expression group, whereas the differences were not significant for the patients with hyperplasia (Figure 7G). We then compared the *ND4/ND1* plasma ratio between the *PGC1A-*low and *PGC1A*-high groups, finding that the *ND4/ND1* ratio was significantly higher in carcinoma than in hyperplasia samples with high *PGC1A* levels, but not for those with low *PGC1A* levels (Figure 7H). As the alterations in the *ND4/ND1* ratio have been shown to be likely derived from mtDNA instability in the tumor, this result further highlights the link between tumor and systemic metabolism, and the relevance of its evaluation for diagnostic purposes.

The number of carcinoma subjects was found to be significantly higher in the high expression group, while the differences were not significant in among the subjects with hyperplasia (Figure 7G). We then compared the levels of the *ND4/ND1* plasma ratio between the *PGC1A-*low and the *PGC1A*-high subjects, and we found that the *ND4/ND1* ratio was significantly higher in carcinoma than in hyperplasia patients with high *PGC1A* levels, but that this was not the case in patients with low *PGC1A* levels (Figure 7H). Since the alterations in the *ND4/ND1* ratio have been shown to be mainly derived from the changes in the tumor, this result further highlights the connection between tumor and systemic metabolism, and the relevance of its evaluation for diagnostic purposes.

## 3. Discussion

As the incidence of thyroid cancer continues to increase, it is ever more important to identify biomarkers for the accurate discrimination of hyperplasia and carcinoma [63]. DNA is a relatively stable molecule that can be simply and reproducibly isolated from blood samples, and is a potentially useful source of biomarkers [64]. Indeed, there are several approved clinical tests for the presence of cancer-associated genomic mutations or epigenetic modifications [16]. These developments, along with the identification of tumor-specific mtDNA mutations, have also spurred interest in the use of mtDNA as a potential biomarker [47]. However, specific applications in this regard have not yet been developed, possibly because most studies do not generally consider the metabolic changes which occur in the tumor or in the organism as a whole.

We aimed to understand how changes in tumor metabolism are related to modifications in mtDNA which can be detected in the tumor and/or in plasma, and how they are related to the systemic metabolic status of patients, and also how they affect mtDNA in non-tumor cells, which would allow us to more adequately test the usefulness of mtDNA for the non-invasive diagnosis of thyroid cancer.

In line with previous observations [65], we found that that the enhanced cell proliferation in tumors is associated with increased cell death and, likely as a result, this led to increased levels of circulating cfDNA. In fact, the levels of cell free genomic DNA (cf-gDNA) were significantly higher in samples from carcinoma than hyperplasia patients. That the enhanced levels gDNA derives mainly from dead cells is supported by the observation the levels of mtDNA present in plasma samples strongly correlated with those of gDNA. However, the total level of cf mtDNA was similar between patients diagnosed with a tumor or a hyperplasia, likely because the mtDNA copy number is lower in thyroid cancer tissue samples than in hyperplasias, which is consistent with similar observations in other cancer types [66]. Nevertheless, as we used crude plasma preparations in the study, the cfDNA detected can be derived not only from bona fide cfDNA but also from damaged PBMCs.

Importantly, the lower abundance of mtDNA in tumors is consistent with the observed increase in markers of reduced mitochondrial activity such altered mitochondrial subcellular distribution [67,68] and activation of AKT [51,69], suggesting a corresponding increased reliance on non-oxidative metabolism in the cancer tissue as compared with hyperplasias. Additionally, the cancer samples showed increased levels of PRX3, a mitochondrial antioxidant that is a surrogate marker of mitochondrial oxidative stress [53] and hence mitochondrial dysfunction, which has also been found to be elevated in some types of cancer [70].

The levels of mtDNA in plasma inversely correlated with those in tumor tissue, suggesting that the lower mtDNA copy number associating with mitochondrial dysfunction was also related to increased cell death rates in the tissue. As this observation could be relevant to help discriminate hyperplasia from carcinoma, we tested the correlation of these parameters in both groups, and while the small sample size precludes us from drawing definite conclusions, it was interesting to note that the observed correlation was maintained in carcinomas but not in hyperplasias when analyzed independently.

We then evaluated the systemic metabolic status of patients through gene expression and protein analysis of relevant metabolic factors in PBMCs, including regulators of oxidative metabolism and mitochondrial biogenesis (PGC-1α and TFAM), PFKFB3, a regulator of glycolytic flux, the mitochondrial antioxidants, PRX3 and MnSOD, and components of the ETC (ATPaseß1, COX IV). At the protein level, we found significantly higher levels of PGC-1α in PBMCs from cancer patients, which was possibly related to the activation of the immune system [71]. However, we were unable to detect significant changes in gene expression of PGC-1α target genes when we compared hyperplasia and carcinoma samples, possibly suggesting that the increase of PGC-1α was not functionally relevant in this context. Aiming to define the possible metabolic differences between the groups, we compared the correlation indices of the tested metabolism genes among them and found a significant difference in the correlations between *PRDX3* and *PFKFB3* between hyperplasia and carcinoma, suggesting that, the normal co-regulation of the response to mitochondrial oxidative stress and the glycolytic flux, which maintains metabolic homeostasis, is disrupted in carcinoma patients. However, the observed differences were not sufficiently robust to fully discriminate the two study groups. We then investigated whether the observed poor response to PGC-1α induction and altered *PRDX3/PFKFB3* correlation, had detectable impact on PBMCs’ mtDNA copy number and found that although the differences among hyperplasias and carcinoma patients did not reach statistical significance (*p*= 0.08), patients with carcinoma subjects had lower levels of mtDNA per cell. This might be indicative of a systemic reduction in mitochondrial activity in carcinoma patients, consistent with the potential loss of PGC-1α activation capacity and altered *PRDX3/PFKFB3* correlation.

Our analysis suggests that tumor metabolic alterations occur in a context of systemic metabolic alterations, a concept which is in line with the well characterized increased cancer risk associated with metabolic diseases [21]. Analysis of the correlations between changes in mtDNA in the thyroid tissue and in PBMCs revealed a significant and positive correlation between mtDNA copy number in thyroid tissue and in PBMCs, further supporting the link between the metabolic status of the tumor and the systemic metabolism. When we analyzed patients with hyperplasia and carcinoma separately, we found that the correlation remained significant only for hyperplasias. Although the small sample size does not allow us to draw definite conclusions, it could be suggestive of the expected increase in metabolic disparity associated with tumor development. We further investigated this by analyzing two mitochondrial genes which have been demonstrated to show differential mutation/deletion ratios in several cancer types, *ND1* and *ND4*. *ND1* strongly correlated with total mtDNA in both thyroid tissue and PBMCs, and the correlation was not significantly different in both contexts. Furthermore, the *ND1*/mtDNA ratio correlated positively and significantly in PBMCs and tissue samples, suggesting it could be indicative of the metabolic coherence between the thyroid and PBMCs in an individual. These results also support the general concept of *ND1* as a low-hit region for mtDNA deletions in cancer [58].

In contrast, *ND4* only correlated with mtDNA in PBMCs, and not in thyroid tissue, and the differences in correlation were statistically significant, suggesting that *ND4*/mtDNA ratios could be specifically altered in the tumor context. This observation is in line with previous data suggesting that *ND4* is a hot spot for mtDNA deletions in cancer [55]. More intriguing was the observation that the *ND4*/mtDNA ratio correlated inversely in thyroid tissue vs. PBMCs.

We evaluated the use of measuring *ND1* and *ND4* in plasma samples, with the caveat that both tissue and PBMC DNA would be present. We first tested the correlation of plasma levels with those found in PBMCs and in thyroid tissue samples. We found that the *ND4*/mtDNA ratio did not correlate in plasma vs. PBMCs, further suggesting that the alteration of this ratio is not relevant nor related to PBMCs. We also we found an inverse correlation between the *ND1*/mtDNA/gDNA ratio in plasma and tissue, which we interpret as an inverse correlation between the preservation of mtDNA in the tissue and its presence in plasma. We next tested the discriminative capacity of these determinations. We found that the *ND1*/gDNA ratio was reduced in carcinomas whereas the *ND4*/gDNA ratio was increased, but the differences observed for these ratios in PBMCs and plasma samples were not significant, limiting their use in the clinical setting. Similarly, the *ND4/ND1* ratio was found to be significantly increased in carcinomas vs. hyperplasias in the thyroid tissue but these differences could not be observed in PBMCs or in plasma samples. Of note, we expected this ratio, often interpreted as an indicator of mtDNA deletion rate, to be lower in carcinomas. However, we found a prior report showing the same observation [29]. Aiming to understand the basis of these differences, we tested the possible differences in correlation for *ND1* vs. *ND4* between carcinoma and hyperplasia samples, and we found that, despite the small sample size, there was a positive correlation in PBMCs between *ND1* and *ND4* in hyperplasia samples which was absent in carcinoma samples, and the differences between the two were significant, further stressing that the observed changes in *ND4* content were of pathological relevance and restricted to cancer patients. Importantly, a similar result was found in our crude plasma samples, although in this case the difference in correlation between the groups did not reach statistical significance (*p*= 0.059). This result, not only supports the concept of a functional connection between the tumor metabolism and systemic metabolism, as we can evaluate it in PBMCs, but also that the *ND4/ND1* contrast analysis could have relevant applications, especially when combining its determination in tumor tissue and in PBMCs or plasma samples.

The functional significance of the changes in *ND4* and *ND1*was further tested by analyzing the differences in correlation between *ND4*/mtDNA or *ND1*/mtDNA values derived from thyroid tissue and PBMCs and relevant genes which signal glycolytic or oxidative metabolism and the response to oxidative stress (*PFKFB3/PRDX3*, *TFAM/PRDX3*, *ATP5BP/PRDX3*), inflammation (*IL1B, IL4*) and angiogenesis (*DLL4*) genes in PBMCs. We found that the changes in the *ND4* ratios correlated better with the changes in metabolic parameters, and with the inflammatory mediator IL-4, which is known to be sensitive to alterations in oxidative metabolism [72], and with the angiogenesis regulator DLL4 in thyroid tissue than in in PBMCs, and the differences were statistically significant in several cases. These results highlight that the changes in the presence of *ND4*, particularly in the tumor tissue are linked to systemic metabolic, inflammatory and angiogenesis related alterations that can be detected in PBMCs. Furthermore, while *ND4* correlated negatively with all the parameters tested, *ND1* was found to correlate positively with the ratio *ATP5BP/PRDX3* in tissue samples, indicating again the contrast in the effects determined by *ND4* and *ND1*, at least in the tissue samples. For PBMCs the only significant correlation identified was for *ND4* vs. *IL1B*, possibly suggesting a particularly relevant link between the presence of *ND4* and the activation of the immune cells in this setting, a concept that would require further evaluation.

Finally, we directly tested the discriminatory power of the combination of the evaluation of the metabolic status as signaled by the gene expression level of *PGC1A* and the plasma *ND4/ND1* ratios. To that end, we first classified the samples based on the levels of *PGC1A* between high and low, and found that the number of patients with carcinomas in the high group was higher than in the low group, whereas in the hyperplasia group the distribution in the groups was not statistically significantly different, indicating that the evaluation of *PGC1A* per se can be functionally significant only in carcinomas. Then, we compared the *ND4/ND1* ratios between the groups and found that, in the *PGC1A-*high group the plasma *ND4/ND1* ratios were significantly higher in carcinomas than in hyperplasias, indicating that the discriminatory capacity of the *ND4/ND1* evaluation using only crude plasma samples can be enhanced by the combined testing of gene expression data.

In the long war against cancer, the unique metabolism of cancer cells has both puzzled and intrigued the scientific community. We are perhaps closer to understanding the fundamental links between metabolism and tumor development, which may open new avenues for both therapy and improved diagnosis. However, the apparently conflicting observations on the role played by mitochondria, respiration, and mitochondrial ROS on cancer development have hampered developments in many cases. We believe the integrative analysis of metabolism in patients, not just restricted to the observation of changes in the tumor context, may really help to pave the way for future developments.

Our study confirms the relevance of local and systemic metabolic control in thyroid tumor development, highlight the links between the metabolic changes occurring in the tumor and the systemic metabolic status of the patients, at least as can be evaluated in PBMCs, and supports the hypothesis that the evaluation of mtDNA status can be a useful tool in patient stratification, especially when data from tissues and PBMCs are combined. Nevertheless, our findings will require further study to validate the biomarkers analyzed here.

It has to be noted, however, that our study has noticeable limitations, which do not allow us to draw definite conclusions on what is the optimal discriminative parameter. This was a single center study, our sample size was small, the number of parameters analyzed were limited, and we used crude plasma samples.

The future development of these approaches would first require a multi-center validation of the findings, followed by the standardization of the analytical procedures. Then, the collected information could be used to train artificial intelligence (AI) and develop final useful diagnostic algorithms of application in the clinic for accurate risk assessment based on the analysis of blood samples. Furthermore, the application of these findings for non-invasive procedures would also be a feasible approach, since wearable and implantable sensors already allow the accurate non-invasive monitoring of parameters related to mitochondrial function such as hearth rate, blood pressure, O_2_ blood saturation and CO_2_ production, which together with the use of trained artificial intelligence (AI) may be used to identify mitochondrial dysfunction related diseases [73]. However, these data would still need to be complemented with analytical methods that actually measure the degree of mitochondrial plasticity loss. Although the development of wearable sensors for a wide variety of molecules is underway, and has been used to detect stress related molecules, the accuracy of these determinations for mitochondrial plasticity remains to be established.

## 4. Materials and Methods

Human subjects. This was a cross-sectional, observational study comprising two independent cohorts of 45 patients and 26 patients. All patients were informed about the study and signed an informed consent form. All patients underwent partial or total thyroidectomy at the Hospital Universitario 12 de Octubre (H12O), Madrid, Spain, because of suspicion of thyroid cancer. Exclusion criteria included presence of comorbidities, confounding pharmacological treatments and patients <16 years or >82 years. Demographic information was gathered from all patients, including sex, age, weight, height and pharmacological treatment history. In all cases, part of the resected thyroid was used for molecular and histological analysis at the Instituto de Investigaciones Biomédicas Alberto Sols (IIBm). Two blood samples were also drawn from patients prior to the surgical procedure for standard biochemical and immunological analysis and to isolate plasma and PBMCs using density-gradient Ficoll-Paque-PLUS separation (GE Healthcare). Following standard histological evaluation by the Pathological Anatomy Unit (H12O), 27 patients of the first cohort were found to have thyroid hyperplasia while 18 had malignant thyroid cancer. In the second cohort, 16 patients had hyperplasia and 10 patients had malignant carcinomas. The study was approved by the Ethics Committees of the IIBm, H12O and the Consejo Superior de Investigaciones Cientificas (SAF2012-37693, CEEHA/2015-0075). The thyroid samples of the two cohorts were processed and analyzed at different time points and by different personnel.

Histology. Tissue samples were first fixed in 4% paraformaldehyde (PFA) in PBS for 24 h. Part of the tissue was then embedded in paraffin and part was first embedded in 30% sucrose in PBS for 48 h and then in OCT compound before freezing at −80 °C. Paraffin (4 μm) and cryostat (15 μm) sections were mounted on slides for subsequent analysis. Hematoxylin and Eosin (H&E) staining was used to analyze thyroid structure and the distribution and size of blood vessels. Images were captured with a bright field microscope (Nikon E90i) equipped with a camera (DS-Q1Mc; Nikon).

Cell proliferation in tissues was evaluated by immunohistochemistry (IHC) analysis of histological sections using antibodies directed against Ki67 (1:100, RM-9106, Thermo Scientific) as previously described [74]. Images were acquired as before and the number of Ki67 positive cells/total number of cells was quantified with the open source Fiji software platform. Subcellular distribution of p-Akt, smooth muscle actin (SMA), VEGFR2 and TOMM22 was analyzed by immunofluorescence (IF) as previously described [74] using the following specific antibodies: p-AKT (Ser473) (D9E) #4060, Cell Signaling Technology, 1:200; VEGFR2 #2479, Cell Signaling Technology, 1:100; SMA-Cy3 CA198 Sigma, 1:100; TOMM22 HPA003037, Merck, 1:200. Samples were mounted in ProLong Mounting reagent (ThermoFisher Scientific) containing DAPI (Invitrogen, dilution 1:500). Images were acquired with a Zeiss LSM 700 or LSM 710 microscope. Fiji software was used to analyze the images. Apoptotic cell death was examined by the Terminal transferase dUTP nick end-labeling (TUNEL) assay using the “In Situ Cell Death Detection Kit, Fluorescein” (Roche) in frozen histological sections. Apoptotic nuclei were counted in a random selection of 10 images per sample acquired with a Nikon E90i fluorescence microscope and analyzed using ImageJ (NIH) software.

Gene expression analysis. Isolated PBMCs or tissue samples were homogenized in the presence of 1ml of TRI Reagent (Invitrogen) and total RNA was isolated following the manufacturer instructions. cDNA was synthesized from total RNA preparations by reverse transcription of 1 μg of RNA using the MMV reverse transcriptase, in a final volume of 20 μL as described in [75]. The mixture was incubated at 37 °C for 45 min and then cooled for 2min at 4 °C. The resulting cDNA was used as template for qPCR analysis using QuantiFast SYBR Green PCR Kit (Qiagen GmbH, Hilden, Germany). Several primers sets used have been previously described [73], and new qPCR oligonucleotides are listed below. Samples were run in triplicates in a Mastercycler ^®^ RealPlex^2^, Eppendorf. *36B4* was used as housekeeping gene.


*PGC1A*


Fw 5′-GGCAGTAGATCCTCTTCAAGATC-3′

Rv 5′-TCACACGGCGCTCTTCAATTG-3′


*PRDX3*


Fw 5′-CCTTTGGATTTCACCTTTGTGTG-3′

Rv 5′-CAAACCACCATTCTTTCTTGGTG-3′


*TFAM*


Fw 5′-GCTTATAGGGCGGAGTGGCA-3′

Rv 5′-AGCTTTTCCTGCGGTGAATCA-3′


*COXIV*


Fw 5′-ACGAGCTCATGAAAGTGTTGTG-3′

Rv 5′-AATGCGATACAACTCGACTTTCTC-3′


*SOD2*


Fw 5′-AGGTTAGATTTAGCCTTATTCCAC-3′

Rv 5′-TTACTTTTTGCAAGCCATGTATCTTTC-3′


*IL1B*


Fw 5′-*GAGCAACAAGTGGTGTTCTC*-3′

Rv 5′-*GATCTACACTCTCCAGCTGT*-3′


*IL4*


Fw 5′-*GCTGCCTCCAAGAACACAAC*-3′

Rv 5′-*TCACAGGACAGGAATTCAAG*-3′


*ATP5BP*


Fw 5′-GTTCCATCCTGTCAGGGACTATG-3′

Rv 5′-TGTGCTCTCACCCAAATGCTGG-3′


*PFKFB33*


Fw 5′-CGTTGCCCAGATCCTGTGGG-3′

Rv 5′-AGCTAAGGCACATTGCTTCCG-3′


*36B4*


Fw 5′-GCGACCTGGAAGTCCAACTA-3′

Rv 5′-ATCTGCTGCATCTGCTTGG-3′

Western blotting. PBMCs and tissue samples were homogenized and whole cell extracts were prepared as previously described [76]. Protein concentration was evaluated by Lowry’s method using the RC/DC Protein Assay (Bio-Rad). A total of 35 μg of protein extract was loaded on 12% SDS-PAGE gels, which was then transferred to Immobilon-P membranes (Millipore). Proteins of interest were identified by western blotting using the specific antibodies listed below, as described [77]. The proteins were visualized using ECL (Amersham) and exposure to photographic films. The films were then scanned and the protein bands were quantitated using the ImageJ software. The original scanned blots are included in the Appendix A.

α-PGC-1α, 101707, Cayman 1:1000.

α-PFKFB3, 13763-1-AP, ProteinTech, 1:1000.

α-PRX3, LF-PA0030, LabFrontier, 1:2000.

α-B-Actin (clone AC-15) 5441, Merck, 1:2000.

Plasma analysis.

Cytokines. The levels of soluble interleukin (IL)-1β, IL-10, IL-4, IL-6 and tumor necrosis factor-alpha (TNFα), were determined in plasma samples by multiplex analysis using ProcartaPlex Immunoassays (ThermoFisher Scientific) by Flow Cytometry at the Centro Nacional de Biotecnología (CNB, CSIC) Flow Cytometry Unit.

Oxidized DNA. The levels of 8-hydroxydeoxyguanosine (8-OHdG) were determined by enzyme-linked immune-absorbant assay (# E-EL-0028, Elabscience).

Antioxidant capacity. Antioxidant capacity was determined using the e-BQC (BioQuoChem) electrochemical analytical system, which measures total (QT), fast (Q1) and slow acting (Q2) antioxidant capacity.

All plasma values were normalized relative to the total protein concentration.

DNA isolation. Total DNA was extracted from plasma, PBMCs and tissue samples using standard phenol-chloroform/isoamyl procedures: 100 µL of plasma was mixed with 500 µL PBS, 6 µL of Triton-X (Carl Roth GmbH + Co KG, Karlsruhe, Germany) and 1 μL of phage λ DNA (250 μg/mL), and was vortexed; 2×10^6^ PBMCs in 30 μL of PBS or 0.5 cm^3^ of tissue were lysed /homogenized in 500 μL of lysis buffer (50 mM Tris HCl pH 8.8, 100 mM NaCl, 10 mM EDTA, 1% SDS and 0.5 mg/mL proteinase K, Roche). The mixtures were vigorously vortexed and then incubated at 55 °C for 30 min (PBMCs) or 2 h (tissue samples). Subsequently, one volume of equilibrated phenol-chloroform-isoamyl alcohol (25:24:1) was added and the mixtures were again vigorously vortexed (Sigma-Aldrich) for 30 s followed by centrifugation at 16.000 × g, for 10 min, room temperature. The upper phase was then transferred to a clean tube for ethanol precipitation overnight at −80 °C. DNA was sedimented by centrifugation 16.000 × g for 1 min at 4 °C. The pellet was extensively washed with 70% ethanol and dried in an open tube at room temperature. Finally, the DNA was resuspended in 20 µL of H_2_O and stored at −20 °C.

DNA analysis. DNA was quantified with a NanoDrop spectrophotometer (Thermo Fisher Scientific, Waltham, MA, USA). Quantification of λ DNA, genomic DNA and mtDNA content was assessed by qPCR using the specific primers listed below. 10 ng of DNA per reaction were amplified using QuantiFast SYBR Green PCR Kit (Qiagen GmbH, Hilden, Germany). Samples were run in triplicates in a Mastercycler ^®^ RealPlex2, Eppendorf. The relative concentration of each DNA was estimated as 2^−ΔΔCt^ where ΔΔCt is calculated as the average Ct for the particular DNA- the average Ct for the phage λ DNA.


*Phage λ*


Fw 5′- GATGGACTTTGGCCAGACC-3′

Rv 5′- TAACACGCTCACCATGAAGC-3′


*Genomic DNA (gDNA), hB2M*


Fw 5′- CACTTTCCACACAGACATCA-3′

Rv 5′- TGGTTAGGCTGGTGTTAGGG-3′


*Mitochondrial DNA (mtDNA), hMito*


Fw 5′- TGTTCCTGCTGGGTAGCTCT-3′

Rv 5′- CCTCCATGATGCTGCTTACA-3′


*ND1*


Fw 5′-ACCCACACCCACCCAAGAAC-3′

Rv 5′- GCGGGTTTTAGGGGTCTTTG-3′


*ND4*


Fw 5′- TGCCTTTAGGTCCAATTATG-3′

Rv 5′-AGGGTGGTTATAGTAGTGTGCATGG-3′


*ND4s*


Fw 5′-TACCTCTTTACGGACTCCACTTATG-3′

Rv 5′-AGGGTGGTTATAGTAGTGTGCATGG-3′

Statistical analysis. ImageJ was used for all image analysis, including: analysis of protein levels of western blot membranes, sub-cellular distribution of mitochondria and AKT, regularity of the vascular tree and number of positive nuclei for Ki67 and TUNEL staining vs. total number of nuclei. Quantitative data were incorporated into Excell spreadsheets. Statistical analysis was performed using Graph Pad Prism 9, SPSS 27 and Psychometrica. Preparation of Graphs was done using Excel and Graph Pad Prism 9. Data in graphs are expressed as mean ± standard deviation (SD). Data normality was evaluated using the Kolmogorov-Smirnov test. Statistical significance of differences between two groups was evaluated by a two-tailed unpaired *t* test; Levene’s test was used for equality of variances. Linear regression and correlation values were determined to compare paired data sets. Comparison of correlations from independent samples was also tested for magnitude (Z) and statistical significance. Values were considered statistically significant at *p* < 0.05. The number of samples included in the analysis is indicated in the figures.

## Figures and Tables

**Figure 1 ijms-24-06453-f001:**
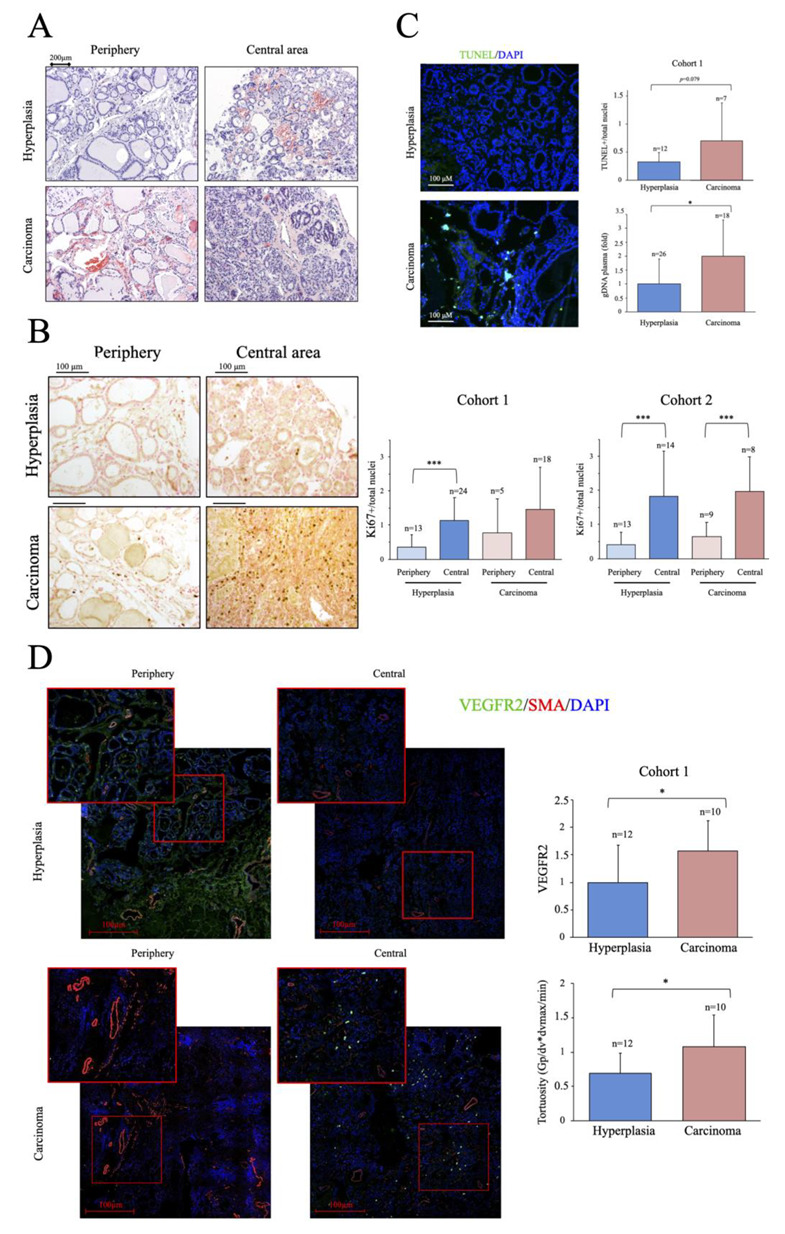
Thyroid cancer samples show higher proliferation and cell death, and a more marked aberrant angiogenesis than hyperplasia samples. (**A**) Hematoxylin and eosin staining (H&E) of histological sections from cancer and hyperplasia samples. (**B**) Immunohistochemistry (IHC) analysis of proliferation using a specific antibody against Ki67. Left panel shows representative images, right panel shows the quantification of images from the two cohorts studied, which indicates the % of positive cells. (**C**) Left panel, representative TUNEL images on histological sections. Top right panel, quantitative analysis of TUNEL assays, which indicates the % of positive cells. Bottom panel, qPCR analysis of gDNA levels in plasma samples. (**D**) Immunofluorescence (IF) analysis of angiogenesis using specific antibodies against SMA and VEGFR2. Left panel, representative images of confocal tile scans (maximal projections) of whole tissue sections, including zooms of relevant areas. Right panels show the quantitative analysis of the confocal images. The top panel shows VEGFR2 levels, indicating the fold change relative to the hyperplasia group, and the bottom panel vascular tortuosity (deviation from circularity), calculated using the formula Gp/dv*dmax/min where *Gp* is the vascular thickness, *dv* is the diameter, *dmax* is the maximal diameter and *dmin* the minimum diameter. Data are mean ± standard deviation (SD). *, *p* < 0.05; ***, *p* ≤ 0.005.

**Figure 2 ijms-24-06453-f002:**
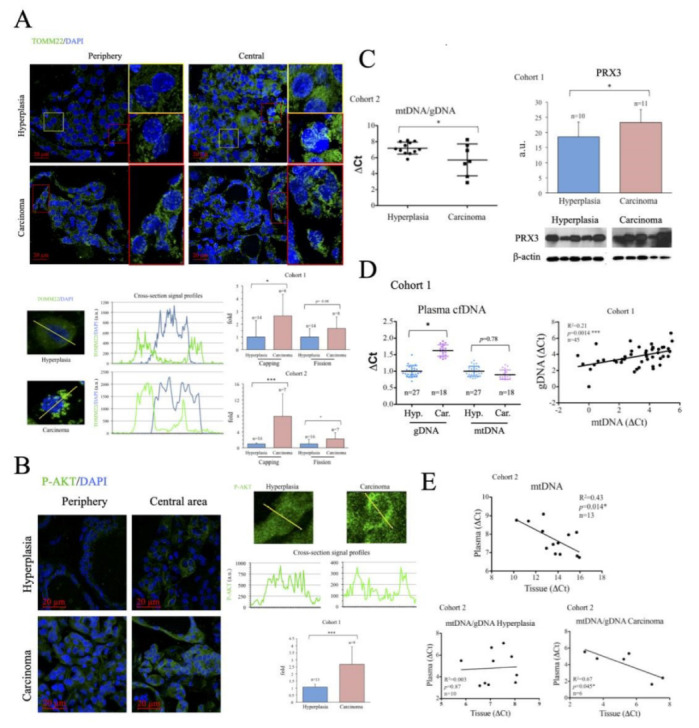
Altered mitochondrial metabolism is more marked in thyroid cancer samples than in hyperplasia samples. (**A**) Immunofluorescence analysis of mitochondrial subcellular distribution using an antibody against TOMM22. Top panel, representative images including zooms of relevant areas. Bottom panels, examples of cross section data acquisition and quantitative analysis. Cross section signal profiles were obtained for TOMM22 (green) and DAPI (blue), the area with DAPI was considered nuclear, the 5 cytosolic adjacent data points were considered peri-nuclear, the ratio of these on both nuclear sided was defined as *capping*. The standard deviation of all the cytosolic data points for TOMM22 *per cell* was used as a surrogate marker of mitochondrial fission. (**B**) Immunofluorescence analysis of p-AKT using a specific antibody. Left panel, representative images. Right panels, examples of cross section data acquisition and quantitative analysis. Cross section signal profiles were obtained for p-AKT (green) and DAPI (blue), the area with DAPI was considered nuclear. The 5 data points closest to the cell edges were considered informative of membrane localization, and the ratio of the membrane values and the average value for all the cytoplasmic signal was calculated for each cell. (**C**) Left panel, qPCR analysis of mtDNA in thyroid tissue. Right panel, thyroid tissue PRX3 western blot data analysis. (**D**) qPCR analysis of mtDNA in plasma samples. Left panel, cfDNA levels in plasma samples. Right panel, correlation and linear regression of mtDNA and gDNA levels in plasma samples. (**E**) Correlation and linear regression of mtDNA levels in plasma and thyroid tissue samples determined by qPCR. Data are mean ± standard deviation (SD). *, *p* < 0.05; ***, *p* ≤ 0.005.

**Figure 3 ijms-24-06453-f003:**
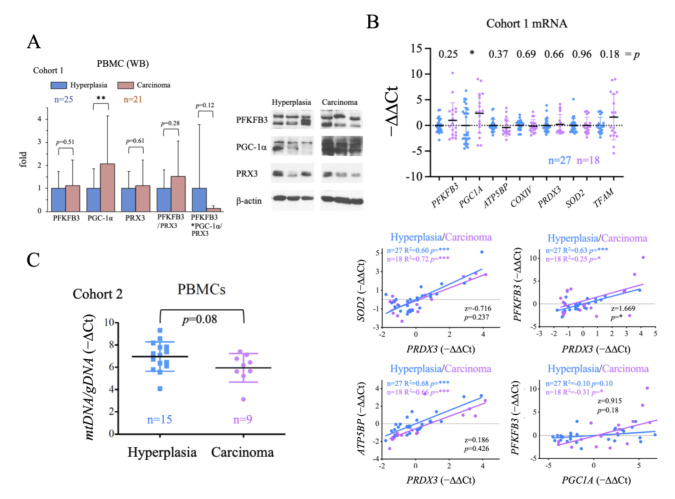
PBMCs from thyroid cancer patients have higher levels of PGC-1α than those from patients with hyperplasia. (**A**) Western blot data analysis of key metabolic regulators (PGC-1α, PFKFB3 and PRX3) in PBMCs. (**B**) qRT-PCR analysis of metabolism-related genes in PBMCs, the top panel shows the −∆∆Ct values and the bottom panel shows the correlation and linear regression of the indicated genes. (**C**) qPCR determination of mtDNA copy number in PBMCs. Data are mean ± standard deviation (SD). *, *p* < 0.05; **, *p* ≤ 0.01, ***, *p* ≤ 0.001.

**Figure 4 ijms-24-06453-f004:**
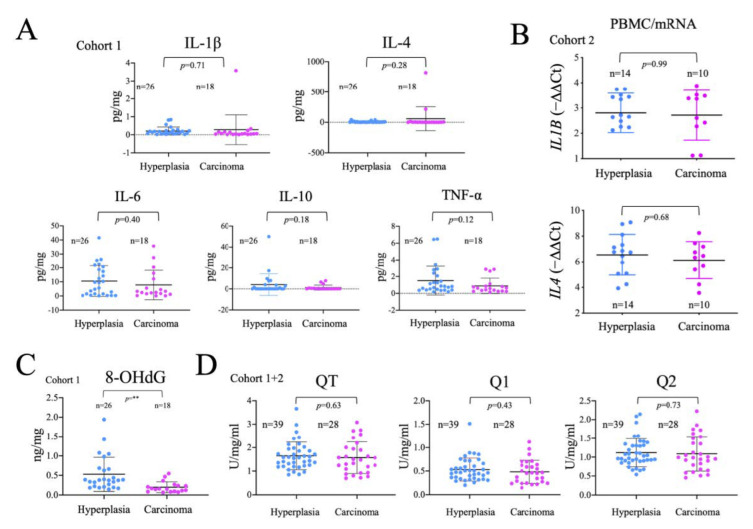
Absence of significant differences in inflammation and antioxidant capacity between patients with cancer and hyperplasia. (**A**) Analysis of inflammatory molecules in plasma samples (IL-1ß, IL-4, IL-6, IL-10, TNFα). (**B**) qRT-PCR gene expression analysis of *IL1B* and *IL4* in PBMCs (**C**) ELISA determination of 8-OHdG in plasma samples. (**D**) Electrochemical analysis of total (QT), fast (Q1) and slow (Q2) antioxidant capacity in plasma samples. Data are mean ± standard deviation (SD). **, *p* ≤ 0.01.

**Figure 5 ijms-24-06453-f005:**
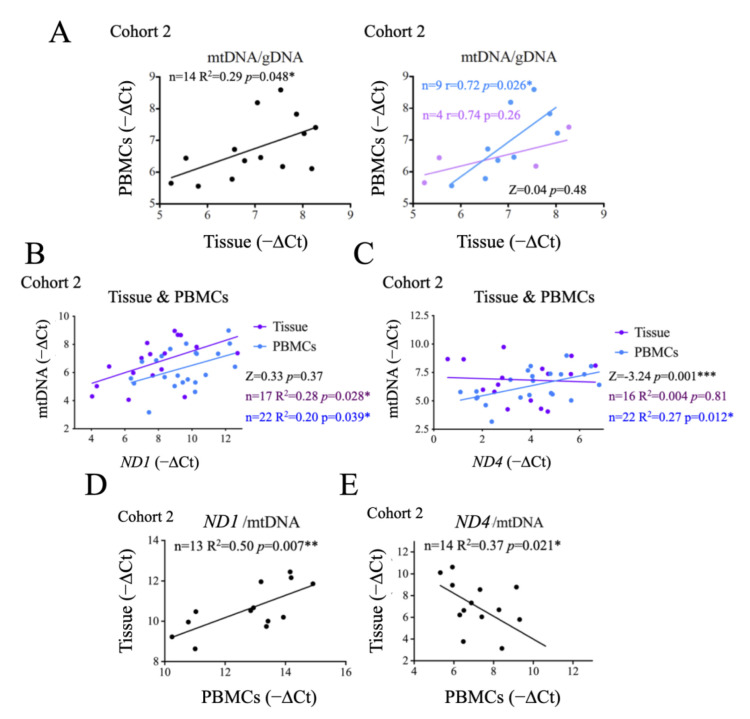
Comparison of mtDNA in thyroid tissue and PBMCs reveals a positive correlation for *ND1* but not for *ND4*. (**A**) Correlation and linear regression analysis of mtDNA in thyroid tissue vs. PBMCs. (**B**) Correlation and linear regression analysis of mtDNA vs. the presence of the mitochondrial gene *ND1*, in thyroid tissue and PBMCs. (**C**) Correlation and linear regression analysis of mtDNA vs. the presence of the mitochondrial gene *ND4*, in thyroid tissue and PBMCs. (**D**) Correlation analysis of *ND1*/mtDNA ratios in thyroid tissue vs. PBMCs. E) Correlation and linear regression analysis of *ND4*/mtDNA in tissue vs. PBMCs. *, *p* < 0.05; **, *p* ≤ 0.01; ***, *p* ≤ 0.005.

**Figure 6 ijms-24-06453-f006:**
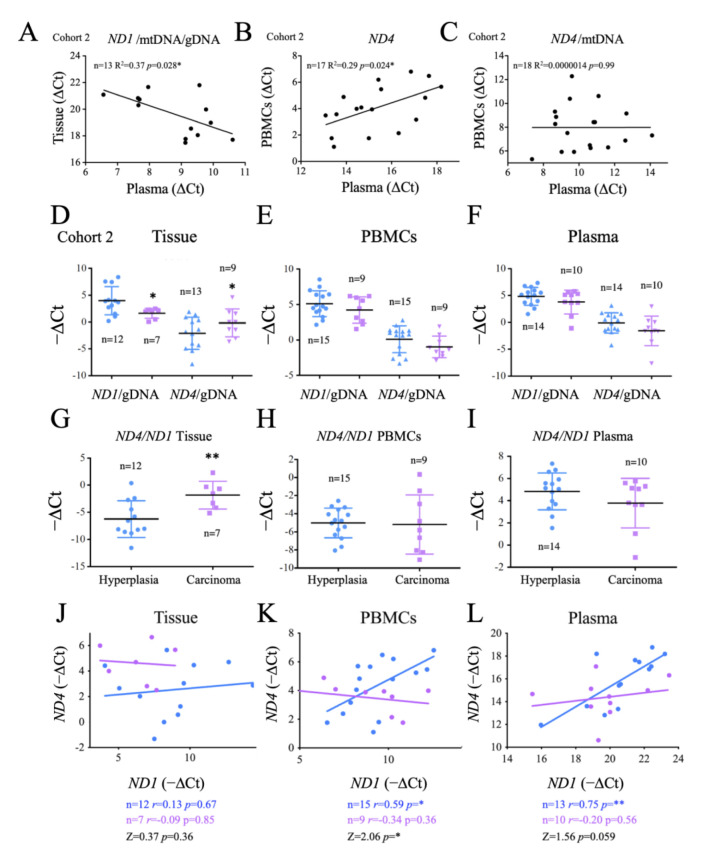
Comparative analysis of mtDNA genes *ND1* and *ND4* in patients with carcinoma and hyperplasia reveals different ratios and correlation indices in thyroid tissues and PBMCs. (**A**) Correlation and linear regression analysis of *ND1*/mtDNA/gDNA in plasma vs. thyroid tissue samples. (**B**) Correlation and linear regression analysis of *ND4* in plasma vs. PBMC samples. (**C**) Correlation and linear regression analysis of *ND4*/mtDNA in plasma vs. PBMC samples. (**D**) Ratios *ND1/*gDNA and *ND4*/gDNA in thyroid tissue samples. (**E**) Ratios of ND1/gDNA and *ND4*/gDNA in PBMC samples. (**F**) Ratios of *ND1*/gDNA and *ND4*/gDNA in plasma samples. (**G**) Ratios of *ND4/ND1* in thyroid tissue samples. (**H**) Ratios of *ND4/ND1* in PBMC samples. (**I**) Ratios of *ND4/ND1* in plasma samples. (**J**) Correlation and linear regression analysis of *ND1* vs. *ND4* in thyroid tissue samples. (K) Correlation analysis of *ND1* vs. *ND4* in PBMCs samples. (**L**) Correlation and linear regression analysis of *ND1* vs. *ND4* in plasma samples. Data are mean ± standard deviation (SD). *, *p* < 0.05; **, *p* ≤ 0.01.

**Figure 7 ijms-24-06453-f007:**
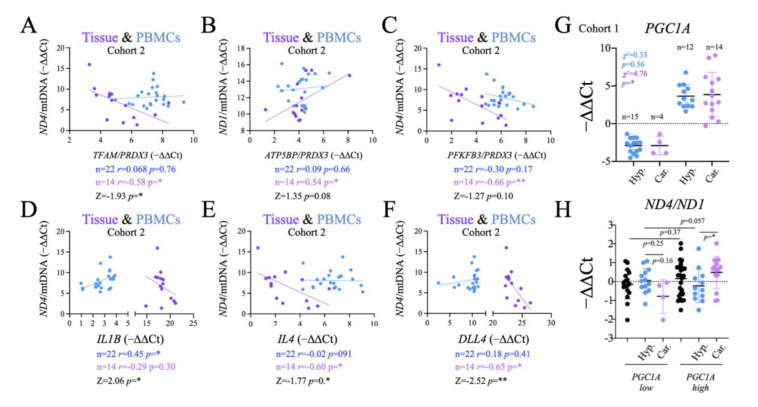
Correlation and linear regression analysis of *ND1*/mtDNA and *ND4*/mtDNA ratios from tissue and PBMC samples with metabolic, inflammation and angiogenesis gene expression data from PBMCs. (**A**) Correlation and linear regression analysis of *ND4*/mtDNA vs. *TFAM/PRDX3* in thyroid tissue and PBMC samples. (**B**) Correlation and linear regression analysis of *ND1*/mtDNA vs. *ATP5BP/PRDX3* in thyroid tissue and PBMC samples. (**C**) Correlation and linear regression analysis of *ND4*/mtDNA vs. *PFKFB3/PRX3* in thyroid tissue and PBMC samples. (**D**) Correlation analysis of *ND4*/mtDNA vs. *IL1B* in thyroid tissue and PBMC samples. (**E**) Correlation and linear regression analysis of *ND4*/mtDNA vs. *IL4* in thyroid tissue and PBMC samples. (**F**) Correlation and linear regression analysis of *ND4*/mtDNA vs. *DLL4* in thyroid tissue and PBMC samples. (**G**) Frequency analysis of *PGC1A*-high vs. *PGC1A*-low patients. (**H**) *ND4/ND1* ratio in *PGC1A*-high vs. *PGC1A*-low patients. Data are mean ± standard deviation (SD). *, *p* < 0.05; **, *p* ≤ 0.01.

**Table 1 ijms-24-06453-t001:** Characteristics of the cohorts studied as classified by the histological diagnosis of the samples provided by the Hospital Universitario 12 de Octubre Pathological Anatomy Unit in accordance with the guidelines of the International Classification of Diseases for Oncology.

	Cohort 1 *n* = 45		*p*	Cohort 2 *n* = 28		*p*
	Hyperplasias *n* = 24	Carcinomas *n* = 21	ns	Hyperplasias *n* = 16	Carcinomas *n* = 12	ns
Male	3	5	ns	3	2	ns
Female	21	16	ns	13	10	ns
Age	62.43+/−12.87	52.76+/−16	0.032	53.06+/−12.80	53.83+/−13.5	ns
BMI	25.72+/−4.49	27.27+/−4.40	ns	29.67+/−7.37	28.76+/−5.10	ns
Fasting glucose (mg/dl)	98.33+/−14.86	93.85+/−10.80	ns	104.13+/−45.43	107.17+/−24.96	ns
Cholesterol (mg/dl)	209.69+/−25.98	206.00+/−43.17	ns	184.64+/−41.09	191.83+/−37.56	ns
Triglycerides (mg/dl)	122.13+/−84.18	116.83+/−67.43	ns	98.77+/−53.40	138.83+/−65.88	ns
Smokers	3	7	ns	2	3	ns
Blood pressure (Max. mm Hg)	137.33+/−18.41	135.50+/−15.58	ns	141.87+/−23.16	127.18+/−28.20	ns
Blood pressure (Min. mm Hg)	78.14+/−10.41	79.70+/−10.53	ns	81.31+/−15.70	79.18+/−11.81	ns

Data are presented as mean +/− standard deviation (SD). *p* < 0.05 was considered statistically signi-ficant. BMI (body mass index).

## Data Availability

Data sets & Original Western Blots are included as Appendix A.

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
