# Peer review of "Evaluation of Mitochondrial Function in Blood Samples Shows Distinct Patterns in Subjects with Thyroid Carcinoma from Those with Hyperplasia"

_ijms, 2023, doi:10.3390/ijms24076453_

Round 1

Reviewer 1 Report

This research investigates metabolic adaptations in thyroid cancer and their correlation with biomarkers of both systemic and tumor-associated metabolic dysfunction. The aim was to better characterize the metabolic characteristics that define thyroid cancer and to identify potential biomarkers with diagnostic value that might be used in minimally invasive procedures. The results show that thyroid tumors present an altered mitochondrial oxidative stress signature detectable in the tissue and in peripheral blood mononuclear cells, that impacts on the presence of mtDNA fragments detectable in plasma samples.  The data from this study suggest that metabolic dysregulation in thyroid cancer can be monitored accurately in blood samples and might be exploited for the accurate discrimination of cancer from hyperplasia.

The manuscript is focused on a relevant topic. It is very well written, presented in a well-structured manner and comprehensive. The aim is clearly defined, the introduction is sufficient and methodology, including statistical analysis, adequate. The results are well presented, figures are easy to interpret and understand, and the conclusions are consistent with the evidence presented. 

Just some minors:

page 20: row 724-725: what were the thyroid cancer subtypes- majority of them were probably papillary thyroid cancer... Where there other such as follicular, medullary... Did the authors try to analyze the metabolic differences between thyroid cancer subtypes in their cohorts?

page 20, row 738: standard abbreviation for immunohistochemistry in English is IHC

page 20, row 740: there are 2 "Scientific"

page 21, rows 771-773 and page 22, rows 846-848: " Each single reaction included 1 µl 846 of DNA (10 ng), 9 µl of QuantiFast SYBR Green PCR Kit (Qiagen GmbH, Hilden, Ger- 847 many) and primers (0.3 µM) to a final volume of 10 µl per reaction" - This is a little bit confusing, I suggest that the exact volumes of either QuantiFast master mix and primers be written.

Author Response

The authors want to thank the reviewers for their comments that, have helped us to significantly improve the manuscript. We have done our best to answer all their suggestions. To facilitate the tracking of the changes we have marked them in blue. 

REVIEWER 1

Q1. Page 20: row 724-725: what were the thyroid cancer subtypes- majority of them were probably papillary thyroid cancer... Where there other such as follicular, medullary... Did the authors try to analyze the metabolic differences between thyroid cancer subtypes in their cohorts?

  1. We analyzed 33 carcinomas, 31 were Papillary Carcinomas, 1 was a Medular Carcinoma and 1a Follicular Carcinoma. Therefore, we did not have enough subjects to try a subtype analysis.

Q2. page 20, row 738: standard abbreviation for immunohistochemistry in English is IHC.

  1. We have now fixed the typo.

Q3. page 20, row 740: there are 2 "Scientific".

  1. We have now fixed the typo.

Q4. page 21, rows 771-773 and page 22, rows 846-848: " Each single reaction included 1 µl of DNA (10 ng), 9 µl of QuantiFast SYBR Green PCR Kit (Qiagen GmbH, Hilden, Germany) and primers (0.3 µM) to a final volume of 10 µl per reaction" - This is a little bit confusing, I suggest that the exact volumes of either QuantiFast master mix and primers be written.

  1. We have changed the text as indicated. It now reads:

771-773. “The resulting cDNA was used as template for qPCR analysis using QuantiFast SYBR Green PCR Kit (Qiagen GmbH, Hilden, Germany). Several primers sets used have been previously described [70], new qPCR oligonucleotides are listed below. Samples were run in triplicates in a Mastercycler ® RealPlex2, Eppendorf. 36B4 was used as loading control.”

846-848. “10 ng of DNA per reaction were amplified using QuantiFast SYBR Green PCR Kit (Qiagen GmbH, Hilden, Germany).”

Reviewer 2 Report

The authors examined the metabolic adaptations in thyroid cancer and their correlation with biomarkers of both systemic and tumor-associated metabolic dysfunction in blood samples.  The data revealed that patients with thyroid tumors present an altered mitochondrial oxidative stress signature detectable in the tissue and peripheral blood mononuclear cells (PBMCs) that impact the presence of mtDNA fragments detectable in plasma samples. Albeit, I consider these findings to provide new insight into cancer-related fields, I still have some suggestions.

1, Most figures and tables are highly professional; however, the authors should guide the readers to the meaning of the images and tables appropriately; otherwise, it is likely to cause misunderstandings. Therefore, I suggest the author consider revising these figures and table legends again. For example, in Fig2, please explain the quantitative analysis data for the green and blue peaks.

2, The author suggests that metabolic dysregulation in thyroid cancer can be monitored accurately in blood samples and might be exploited for the accurate discrimination of cancer from hyperplasia. However, It would be much better if the authors could provide some Workflow or Scheme for this research, I suggest that they can take a look at the recent paper in MDPI (PMID: 35328243, 24619302 , 34834441)

3, In table 1, the author presented the characterization of the cohort: basic metabolic risk factors and histological analysis. Since most P value was not statistically significant, I suggest they may validate their data via the public database, such as UK Biobank or NCBI GEO…etc.

4, In Fig 1-3, the author presented the bar graph of statistical analysis, please explain why you used Kolmogorov-Smirnov test for the current study. The author may need to use other statistical analyses, such as ANOVA to calculate the P-value for three or more groups of data, and please update the “Statistical Analysis” of the Method during further revision(PMID: 36765669, 36555654, 32064155)

5, There are few typo issues for the authors to pay attention to; please also unify the writing of scientific terms. “Italic, capital”? Please also add scale bar for Figure1-2, for example, the scale bar of immunofluorescence data was not clear.

Author Response

The authors want to thank the reviewers for their comments that, have helped us to significantly improve the manuscript. We have done our best to answer all their suggestions. To facilitate the tracking of the changes we have marked them in blue. 

REVIEWER 2

Q1. The authors should guide the readers to the meaning of the images and tables appropriately. Therefore, I suggest the author consider revising these figures and table legends again. In Fig. 2, please explain the quantitative analysis data for the green and blue peaks.

  1. The figure legends have been modified aiming to increase clarity. In particular, the Fig.2 legend has been modified as follows:
  2. A) Immunofluorescence analysis of mitochondrial subcellular distribution using an antibody against TOMM22. Top panel, representative images including zooms of relevant areas. Bottom panels, examples of cross section data acquisition and quantitative analysis. Cross section signal profiles were obtained for TOMM22 (green) and DAPI (blue), the area with DAPI was considered nuclear, the 5 cytosolic adjacent data points were considered peri-nuclear, the ratio of these on both nuclear sided was defined as Capping. The standard deviation of all the cytosolic data points for TOMM22 per cell was used as a surrogate marker of mitochondrial fission. B) Immunofluorescence analysis of p-AKT using a specific antibody. Left panel, representative images. Right panels, examples of cross section data acquisition and quantitative analysis. Cross section signal profiles were obtained for p-AKT (green) and DAPI (blue), the area with DAPI was considered nuclear. The 5 data points closest to the cell edges were considered informative of membrane localization, and the ratio of the membrane values and the average value for all the cytoplasmic signal was calculated for each cell.

Q2. The author suggests that metabolic dysregulation in thyroid cancer can be monitored accurately in blood samples and might be exploited for the accurate discrimination of cancer from hyperplasia. However, It would be much better if the authors could provide some Workflow or Scheme for this research, they can take a look at the recent papers in MDPI (PMID: 35328243, 24619302, 34834441).

  1. We have now included the following text in the discussion as an outline of a possible future Workflow.

“The future development of these approaches would first require a multi-center validation of the findings, followed by the standardization of the analytical procedures. Then, the collected information could be used to train artificial intelligence (AI) and develop final useful diagnostic algorithms of application in the clinic for accurate risk assessment based on the analysis of blood samples. Furthermore, the application of these findings for non-invasive procedures would also be a feasible approach, since wearable and implantable sensors already allow the accurate life non-invasive monitoring of parameters related to mitochondrial function such as hearth rate, blood pressure, O2 blood saturation, CO2 production, that together with the use of trained artificial intelligence (AI) may be used to identify mitochondrial dysfunction related diseases (PMID: 35328243). However, these data would still need to be complemented with analytical methods that actually measure the degree of mitochondrial plasticity loss. Although the development of wearable sensors for a wide variety of molecules is underway, and has been used to detect stress related molecules, the accuracy of these determinations for mitochondrial plasticity remains to be stablished”. 

Q3. In table 1, the author presented the characterization of the cohort: basic metabolic risk factors and histological analysis. Since most p values were not statistically significant, I suggest they may validate their data via the public database, such as UK Biobank or NCBI GEO…etc.

  1. To clarify this point we have now included in the following texts and references:

“A recent epidemiological study based in Korea with 173,343 participants showed that metabolically unhealthy women, either normal weight or obese, had an increased risk of thyroid cancer [HR (95% CI) = 1.57 (1.02–2.40) and 1.71 (1.21–2.41), respectively) compared with non-obese women without metabolic abnormalities. However, this significant association was not observed in men. Thyroid cancer risk was higher among non-obese women with high Waist Circumference (WC) [≥85 cm; HR (95% CI) = 1.62 (1.03–2.56)] than in non-obese women with low WC, and in obese women with low HDL-cholesterol [<50 mg/dL; HR (95% CI) = 1.75 (1.26–2.42)] compared with non-obese women with high HDL-cholesterol (30).”

“Nevertheless, the most common histological staining used by pathologists for thyroid cancer diagnosis is still H&E staining (31).”

Q4. In Fig 1-3, please explain why you used Kolmogorov-Smirnov test. The author may need to use other statistical analyses, such as ANOVA to calculate the P-value for three or more groups of data, and please update the “Statistical Analysis” of the Method during further revision.

  1. The Kolmogorov-Smirnov test was used verify the normal distribution of the data. We did not use ANOVA but just t test because we calculated p values for the comparison of just two groups of data.
